# Impact of Cytomegalovirus Infection and Genetic Background on the Frequencies of Peripheral Blood Suppressor Cells in Human Twins

**DOI:** 10.3390/pathogens10080963

**Published:** 2021-07-30

**Authors:** David Goldeck, Lisbeth Aagaard Larsen, Kaare Christensen, Klaus Hamprecht, Lilly Öttinger, Karin Hähnel, Graham Pawelec

**Affiliations:** 1Independent Researcher, Kidlington OX5 2PB, UK; 2The Danish Twin Registry, University of Southern Denmark, 5000 Odense C, Denmark; laalarsen@health.sdu.dk (L.A.L.); KChristensen@health.sdu.dk (K.C.); 3Department of Clinical Genetics and Department of Clinical Biochemistry and Pharmacology, Odense University Hospital, 5000 Odense C, Denmark; 4Institute of Medical Virology, University of Tübingen, D-72072 Tübingen, Germany; klaus.hamprecht@med.uni-tuebingen.de; 5Department of Psychiatry and Psychotherapy, Section for Dementia Research, University of Tübingen, D-72072 Tübingen, Germany; lilly.oettinger@gmx.de; 6Department of Internal Medicine II, University Hospital Tübingen, D-72072 Tübingen, Germany; karin.haehnel@uni-tuebingen.de; 7Department of Immunology, University of Tübingen, D-72072 Tübingen, Germany; 8Health Sciences North Research Institute, Sudbury, ON P3E 2H3, Canada

**Keywords:** cytomegalovirus, regulatory T cells, myeloid-derived suppressor cells, twins, heritability, genetics

## Abstract

Frequencies and proportions of regulatory T cells (Tregs) and myeloid-derived suppressor cells (MDSCs) in peripheral blood may be informative biomarkers for certain disease states. The influence of genetics and lifetime pathogen exposures on Treg and MDSC frequencies is largely unexplored. Cytomegalovirus (CMV) establishes a latent infection and causes an accumulation of late-differentiated CD8+ memory T cells, commonly associated with a lower frequency of naive cells. Here, analyzing peripheral blood mononuclear cells by multicolor flow cytometry, we found a tendency towards lower frequencies of CD4+CD25+FoxP3+ Tregs in CMV-seropositive than -seronegative middle-aged individuals (*p* = 0.054), whereas frequencies of lineage-negative CD14+HLA-DR-MDSCs were significantly lower in CMV-seropositive participants (*p* = 0.005). Assessing associations with the presence of antibodies against different CMV structural proteins, rather than merely assigning seropositivity or seronegativity, failed to yield any closer associations. Examining Treg subsets revealed at most a minor role of the individual’s genetic background, based on an analysis of monozygotic (MZ, *n* = 42) versus dizygotic (DZ, *n* = 39) twin pairs from the Danish Twin Registry. The same was true for MDSCs. These initial results suggest that an immunological history of exposures is more important than genetics in determining overall human suppressor cell levels.

## 1. Introduction

There is clear influence of inter-individual heterogeneity on immune capacity, response to infection and general health status in humans, but relative contributions of genetic background-vs.-lifetime pathogen exposures are unclear. Differences in the frequencies of peripheral blood regulatory T cells (Tregs) and/or myeloid-derived suppressor cells (MDSCs) have been associated with certain disease states, but it is not known whether or which pathogen exposures may influence their status and what impact the genetic background may have. One factor which clearly markedly influences the frequencies and proportions of peripheral T cells with early- and late-differentiated phenotypes is seropositivity for Cytomegalovirus (CMV) [1], but its influence on Tregs and MDSCs is not well-established. CMV is a widespread beta herpesvirus, a chronic infection which is usually asymptomatic in immunocompetent hosts. However, CMV is a serious pathogen in immunocompromised patients, for example as organ transplant recipients or AIDS patients [2,3,4]. Chronic CMV infection is reflected in the presence of antibodies to different viral structural proteins, the balance of which can change following the periodic reactivation of the virus. This can be triggered by a variety of different events including infection, inflammation or stress. Preventing and/or controlling reactivation requires immunological resources and an investment in defense reflected in the increased proportions of CMV-specific cells and antibody titers [5]. There is a great deal of inter-individual variability in the manner in which the distribution of T cell phenotypes and their proportions, as well as the distribution of antibodies against different CMV molecules. A range of variables may influence this, such as age at primary infection, duration of latency or rounds of reactivation, but the genetic background may also be important in this respect. We have previously obtained circumstantial evidence for this from the Leiden Longevity Study (LLS) of familial longevity [6], but whether genetic heterogeneity plays a role in determining the frequencies of important T cell subsets such as Tregs in CMV-infected versus uninfected adults is not known and is difficult to establish in humans. One of the limited avenues available to explore this issue is to examine monozygotic (MZ) vs. dizygotic (DZ) twin pairs on the assumption that the former is essentially identical genetically whereas the latter are no more similar than other sibling pairs which share the same environment and exposures, at least early in life. Using this approach, we previously established a significant intraclass correlation between cotwins of MZ, but not DZ pairs, for the differentiation status of CD4+ and CD8+ T cell subsets. Classical heritability analysis confirmed a substantial contribution of genetics to the differentiation status of T cells in CMV infection, but Tregs were not examined in that study [7].

## 2. Results

### 2.1. Basic Study Cohort Characteristics

Eighty-one same-sex twin pairs participated in the study, the majority being 41–77-year-old women (Appendix A). CMV serostatus was determined for all study participants and twin pairs were grouped according to their CMV serostatus for further analysis: 16 pairs were concordant CMV-seronegative and 21 were CMV discordant. These two groups were used for the analysis of the impact of CMV-seropositivity on immune phenotypes. And further 44 concordant CMV-seropositive pairs were included in the correlation analysis.

To investigate a potential impact of age we divided the study participants into two subgroups, one with age higher than the median of 48 years and one lower or equal to the median. Then we compared frequencies of immune cell subsets between these groups but found few differences. Similarly, we grouped the twins according to their sex, and found little difference except slightly higher frequencies of Lin-CD16-CD14- cells (presumably dendritic cells) in male compared to female study participants. As a result, these two parameters age and sex were not considered in further analyses.

### 2.2. Effect of CMV-Seropositivity on Peripheral Treg and MDSC Frequencies

First, we determined the impact of CMV on frequencies of peripheral Treg and MDSC phenotypes in this cohort, analyzing all participants together. For this, individuals were grouped solely according to CMV-serostatus and the proportion of T cells expressing the markers CD25 and FoxP3 for Treg analysis, and the frequencies of Lin-CD14+HLA-DR- cells for MDSCs were quantified. Frequencies of both tended to be lower in CMV-seropositive than -seronegative individuals (Figure 1). A great deal of inter-individual heterogeneity is seen in this small cohort and only the difference between CD4+ CD25+ FoxP3+ T cell in CMV-seropositive-vs-seronegative individuals was even borderline significant (*p* = 0.0536). The frequencies of subsets of Tregs within populations of CD4+ CD25+ FoxP3+ T cells, representing induced (Helios+), resting (CD45RA+FoxP3low), activated (CD45RA-FoxP3high) and potentially non-suppressive Tregs (CD45RA-FoxP3low) did not reveal any impact of CMV infection (Figure 1a). Similarly, no differences could be discerned between CMV-seropositive and -seronegative participants regarding frequencies of FoxP3+ CD8+ T cells (Figure 1b). However, CMV-seropositive individuals exhibited significantly lower frequencies of lin-CD14+HLA-DR- MDSCs than -seronegative donors (*p* = 0.0046), again with a great deal of inter-individual variation (Figure 1c).

For the three populations of suppressor cells which showed the greatest difference between CMV-seronegative and CMV-seropositive individuals, we performed a more detailed analysis (Figure 2). We divided the CMV-seropositive study participants according to their antibody reactivity to each of 6 CMV peptides tested individually; all possessed antibodies binding p150 and gB1, so these were excluded from this analysis, which was then performed for IE-1, CM2, p65 and gB2. Frequencies of CD25+FoxP3+, CD25+FoxP3+Helios+ and CD14+HLA-DR- cells were compared between seronegative and seropositive individuals and no significant difference was found except for a slightly higher frequency of CD14+HLA-DR- cells in p65- compared to p65+ twins (*p* = 0.045, Figure 2c). In addition, all twins were grouped according to the number of different CMV molecules their antibodies bound (summing the number of responses to each component), but again, no differences in Treg frequencies were observed. For CD14+HLA-DR- MDSCs a trend towards lower frequencies with higher number of reactions was seen (Appendix A).

### 2.3. Correlations of Treg and MDSC Frequencies within Monozygotic and Dizygotic Twins 

Considering the large inter-individual variability, as expected in humans and commonly reported for many of the parameters analyzed here, especially in the CMV-seropositive group, we asked whether co-twins had correlated suppressor cell frequencies. For this, twin pairs were grouped into concordant negative (both twins CMV-seronegative), concordant positive (both twins seropositive) and discordant (only one twin seropositive) and an intraclass correlation analysis was performed. Due to the high frequency of CMV infection, only very few pairs were CMV-seronegative or discordant (Appendix A). Therefore, we report only concordant CMV-seropositive twins. We observed a statistically significant intraclass correlation in both MZ and DZ twins for the frequency of CD25+ FoxP3+ CD4+ Tregs (Table 1, Appendix A). However, for certain composite phenotypes, there was a statistically significant correlation between MZ but not DZ twins. This was most striking for the frequencies of resting Tregs, suggesting a genetic influence on regulating the maintenance of a pool of available Tregs. Reciprocally, there were no correlations for induced Tregs in MZ twins (Table 1, Appendix A). Similarly, we observed a substantial or strong intraclass correlation between both MZ and DZ co-twins for frequencies of MDSCs defined as lin-CD14+HLA-DR- cells. However, the frequencies of “classical” monocytes (CD14+HLA-DR+) were correlated only in MZ but not DZ pairs (Table 1, Appendix A) An impact of age was not detected (Appendix A).

## 3. Discussion

In previous studies we reported a substantial contribution of the genetic background to the differentiation status of peripheral blood αß and γδ T cells and a potential contribution to resistance to the effects of CMV infection on the distribution of T cell phenotypes in humans. These studies revealed an intraclass correlation of T cell frequencies in MZ twins which was weaker or absent in DZ twins [7], and a lesser impact of CMV in the offspring of long-lived parents in family studies [6]. The aim of the current study was to extend this analysis to include populations of Tregs and MDSCs and to determine the impact of CMV infection in MZ-vs-DZ twins.

Chronic low-grade inflammation is considered a hallmark of aging that contributes to organ damage and the diseases of aging [8]. This is paralleled by a functional remodeling of the immune system which may manifest in deleterious functionality [9]. Part of this remodeling is likely to involve responses to the heightened inflammatory status dubbed “inflammageing” [10], both in terms of innate immune cells, e.g., MDSCs [11] and adaptive immune cells, e.g., regulatory T cells [12]. The mechanisms resulting in the state of “inflammageing” are manifold and reflect a balance of pro- and anti-inflammatory effectors [13] influenced by the response to external challenges and the presence of increasing organismal cellular senescence [14]. A major challenge for the human immune system arises from the necessity of controlling latent infection with CMV [15]. The overall impact of CMV infection on human health remains controversial, with estimates ranging from a marked negative effect on longevity [16] to little or no effect [17]. Given the potential contribution of CMV to inflammageing via its impact on the immune system, one possible explanation for its different effects in different populations might be genetic, as hinted at in studies of familial longevity [6]. Because increased levels of MDSCs and Tregs as a result of inflammageing may contribute to increased severity of infectious disease and potentially cancer in older adults, genetic control of these mechanisms would be expected to impact on overall health and longevity. Hence, in order to investigate a possible influence of heritability on Treg and MDSC levels in humans, here were assessed their frequencies in MZ-vs-DZ twin pairs in a pilot study. 

However, our null hypothesis that CMV infection would markedly influence the frequencies of Tregs in peripheral blood, as it does the frequencies of many other T cell subsets, was disproven (although the frequency of CD4+ CD25+ FoxP3+ T cells did tend to be higher in CMV-seronegative individuals, Figure 1a). In contrast, the frequency of lin-CD14+HLA-DR- MDSCs differed significantly between CMV-seropositive and -seronegative donors, with average frequencies of MDSCs lower in CMV-seropositive donors (Figure 1c). These results would not be expected were CMV exerting a pro-inflammatory effect or directly infecting and stimulating myeloid cells. Most studies of this type limit comparisons to CMV-seronegative-vs-seropositive donors without taking the heterogeneity of the antibody responses into account. Here, we separately assessed the impact of the presence of antibodies for different CMV structural proteins among CMV-seropositive individuals (Figure 2) but failed to increase the robustness of any of the measured associations by these means.

Consistent with results from studies of familial longevity [6], comparing infection rates in MZ-vs-DZ, twin pairs suggested that the probability of becoming infected with CMV was negligibly influenced by genetic background. This would be expected for infection with an ancient co-evolved virus which infects essential all individuals in the population in low-income countries and a large fraction increasing with age in high-income countries. We found an intraclass correlation in both MZ and DZ twin pairs for the overall frequency of CD25+ FoxP3+ CD4+ Tregs in CMV-seropositive twin pairs, contrary to our null hypothesis that levels of Tregs would be more closely correlated in MZ than in DZ twin pairs. However, a significant difference did emerge when considering resting Tregs, which were strongly correlated in MZ but not DZ twin pairs. The biological meaning of genetic control of resting Treg frequencies is open to speculation but has ramifications for understanding differences in immune homeostasis in different individuals. Similarly, there were few differences between MZ and DZ twins regarding overall frequencies of MDSCs, whereas classical monocytes (CD14+HLA-DR+) correlated only in MZ twin pairs. 

This pilot study on very limited numbers of twin pairs (*n* = 81), which for logistical reasons was challenging and expensive to perform, was not able to detect a major influence of genetic background or CMV infection on most of the many composite regulatory T cell and myeloid cell subsets measured. Hence, it was decided not to expand the study by extending the Treg and MDSC subset phenotypes investigated as it was considered that the prospects for extracting more meaningful data would be limited. These negative findings do, however, raise the intriguing questions of why CMV has such a marked effect on the distribution of some immune cell subsets, especially CD8+ T cell subsets (important effector arm of adaptive immunity) but not so much on regulatory components, and why a genetic influence is discernible on effector T cell subsets but also not on regulatory immune cells.

How do our present results compare to other twin studies? As alluded to, these are few and far between, but an extensive major study in 105 twin pairs some years ago concluded that variation in the human immune system was largely caused by external exposures and that in MZ twin pairs discordant for CMV, many immune parameters were affected [18]. That study did find that the frequency of Tregs was strongly correlated in MZ twin pairs, but only in younger, not older, twins. Our data reported here are thus consistent with this finding, as we were only able to examine middle-aged twins. A more recent study from the same group using different markers and analytical techniques has confirmed this general picture for Tregs but did not explore Treg subset distribution [19]. On the other hand, another recent study on 497 twins concluded a high degree of heritability for Tregs, and less for monocytes, but again used different markers to identify them, and only female twins were studied [20]. Hence, differences in techniques and the paucity of twin studies make it difficult to reach any firm conclusions regarding heritability of traits related to immune regulatory circuits in humans.

## 4. Materials and Methods

### 4.1. Study Population

The study cohort was recruited from the 2008–2011 survey of middle-aged Danish twins undertaken by the Danish Twin Registry [21]. A total of 81 complete twin pairs (162 twins) were randomly selected among the 14,000 participating twins born 1931–1969. Only complete same-sex twin pairs were included (see Appendix A). Zygosity was established through a questionnaire on the degree of similarity between twins in a pair [22]. A convenience sample of 134 twins (67 twin pairs) was selected for analyzing Tregs and 136 twins (68 twin pairs) for MDSC analysis. Whole blood was collected in Vacutainer CPT tubes. PBMCs were isolated immediately, according to the manufacturer’s protocol (BD Vacutainer^®^ CPT™, REF 362761, Heidelberg, Germany) and cryopreserved. Aliquots of PBMCs were stored at −80 °C for 24 h and then transferred to liquid nitrogen for long-term storage. This study was approved by The Science Ethics Committee of Southern Denmark (project number S-VF-19980072).

### 4.2. Flow Cytometry

All experiments were performed using established protocols, as in our previous studies [7,10]. All staining steps were performed in PFEA buffer (PBS, 2% FCS, 2 mM EDTA, and 0.01% azide). After thawing, PBMCs were treated with human Immunoglobulin, GAMUNEX (Bayer, Leverkusen, Germany), and ethidium monoazide (EMA) (Invitrogen, Karlsruhe, Germany) for 10 min on ice to block Fc receptors and label nonviable cells. Cells were first stained with conjugated monoclonal antibodies for 20 min at 4 °C to detect surface markers, followed by fixation and permeabilization with BD Fix/Perm and intracellular staining with anti CD3-PacificOrange, Helios-FITC and FoxP3-Alexa647 for another 30 min. Treg panel surface marker antibodies were CD4-PerCP, CD8-APC-H7, CD25-PE, CD45RA-BV421, CD45RO-PE-Cy7 and for MDSC characterization CD3-BV605, CD19-BV605, CD56-BV605, CD11b-APC-Cy7, CD14-PE-Cy7, CD15-FITC, CD16-PacificBlue, CD103-BV711, and CD124-PE HLA-DR-PerCP-Cy5.5. After incubation, cells were washed and analyzed on an LSR II cytometer with FACSDiva software (BD Biosciences, Heidelberg, Germany). The spectral overlap between all channels was calculated automatically by the BD FACSDiva software, after measuring negative and single-color controls with BD comp beads. Data were analyzed using FlowJo software (BD Biosciences).

For data analysis, our standard procedures [23] were applied to gate lymphocytes in a forward light scatter versus side scatter dot plot according to their size and granularity. After excluding the doublets in an FSC-area vs. FSC-width and SSC-area vs. SSC-width plots, EMA-negative viable cells were selected. T cells within the viable gate were characterized as CD3+ cells, followed by gating CD4+ and CD8+ cells in a CD4 vs. CD8 dot plot. T cell activation status was determined based on the expression level of CD25 and Tregs by expression of FoxP3. The phenotype of Tregs was characterized by Helios expression (induced Tregs) and by plotting FoxP3 against CD45RA to determine resting (CD45RA+FoxP3low), activated (CD45RA-FoxP3high) and non-suppressive (CD45RA-FoxP3low) Tregs (Appendix A). In a parallel approach with a second panel the frequencies of lineage negative (lin-: CD3-CD19-CD56-) classical (CD14+CD16-), intermediate (CD14+CD16+) and non-classical (CD14lowCD16+) monocytes was analyzed out of the total living (EMA-) cells. MDSC analysis was conducted by plotting HLA-DR expression against CD14. Statistical analysis used Graph Pad Prism 5.

### 4.3. CMV-Serology

CMV IgG serostatus was determined by means of a recombinant CMV IgG immunoblot kit (Mikrogen, Neuried, Germany) using six different viral target molecules (IE-1, p150, CM2, p65, gB1, and gB2). An individual was defined as CMV-seropositive if their serum bound p150 or other antigens, relative to a control, according to the information of the manufacturer, as described previously [1].

## Figures and Tables

**Figure 1 pathogens-10-00963-f001:**
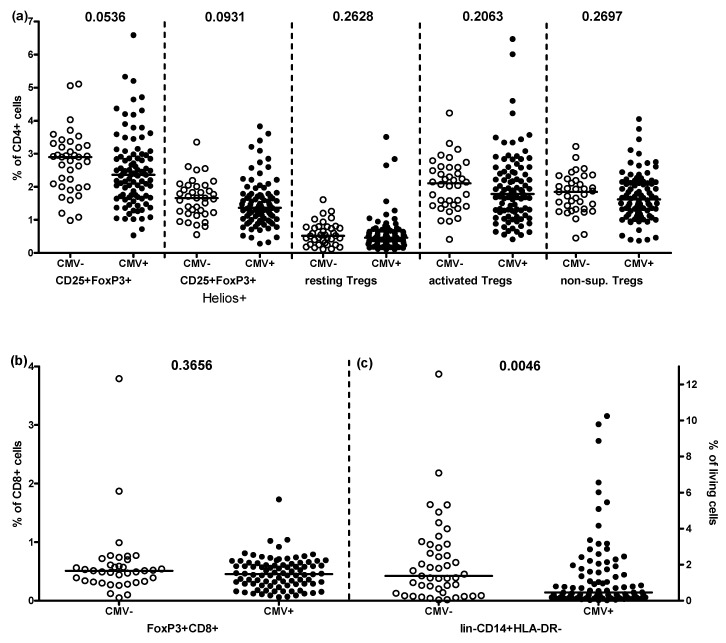
Impact of CMV-seropositivity on the frequency of suppressor cells within twins. (**a**) From left-to-right: frequencies of cells expressing CD25 and FoxP3 in CMV-seronegative and CMV-seropositive participants, then those additionally expressing Helios, then resting CD45RA+FoxP3low Tregs, next activated CD45RA-FoxP3high Tregs, and finally CD45RA-FoxP3low potentially non-suppressive Tregs. (**b**) Frequencies of FoxP3+ CD8+ T cells in CMV-seronegative and CMV-seropositive participants. (**c**) Frequencies of lin-CD14+HLA-DR- MDSCs. Each symbol represents one donor. Horizontal bars represent the median of the group. *p* values were calculated using the Mann Whitney U test with no need for Bonferroni correction.

**Figure 2 pathogens-10-00963-f002:**
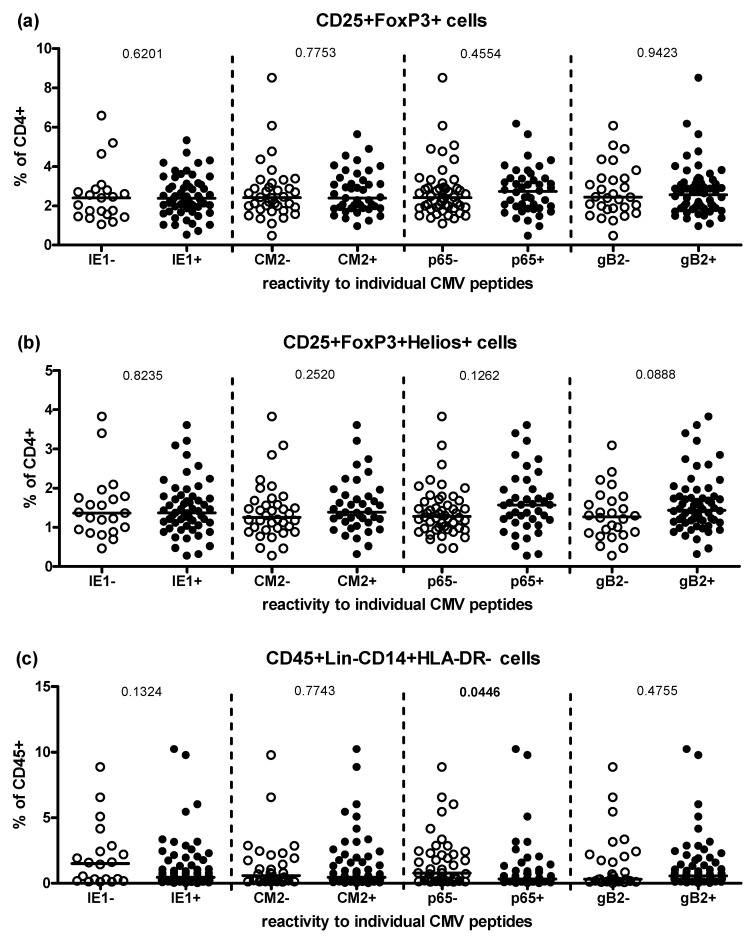
Impact of seropositivity to different CMV peptides on the frequency of suppressor cells within CMV-seropositive twins. Of the six analyzed CMV peptides p150 and gB1 were excluded here, because all CMV-seropositive study participants had antibodies binding these two molecules. Borderline positive individuals were excluded for each peptide. Each symbol represents one donor. Frequencies of CD25+FoxP3+ (**a**), CD25+FoxP3+Helios+ (**b**) and CD14+HLA-DR- cells (**c**) were compared between seronegative and seropositive individuals with Mann Whitney *U* test. The horizontal bars represent the median of each group.

**Table 1 pathogens-10-00963-t001:** Correlation of immune phenotype frequencies within CMV-seropositive twin pairs.

Immune Subset	MZ Nonparametric ^1^			MZ Linear R. ^2^		DZ Nonparametroic ^1^			DZ Linear R. ^2^	
	Sr	*p* Value	Outlier	r^2^	*p* Value			Outlier	r^2^	*p* Value
Tregs, 19 MZ and 17 DZ pair										
CD25+	0.42	0.077	0	0.21	0.059	0.59	0.012	0	0.43	0.005
FoxP3+	0.53	0.029	0	0.17	0.098	0.69	0.002	0	0.73	<0.001
CD25+FoxP3+	0.60	0.009	0	0.29	0.022	0.64	0.006	0	0.69	<0.001
Non suppressive Tregs	0.46	0.062	0	0.24	0.045	0.635	0.006	0	0.38	0.008
CD25+FoxP3+Helios+	0.40	0.11	0	0.18	0.092	0.62	0.008	0	0.58	<0.001
resting Tregs	0.71	0.001	0	0.93	<0.001	−0.19	0.46	0	0.13	0.154
activated Tregs	0.55	0.014	0	0.60	<0.001	0.58	0.014	0	0.57	0.001
activated CD25+ Tregs	0.51	0.004	0	0.21	0.063	0.64	0.006	0	0.57	0.001
FoxP3-Helios+	0.78	<0.001	0	0.59	<0.001	0.64	0.005	0	0.20	0.075
FoxP3+Helios+	0.28	0.271	0	0.20	0.076	0.66	0.004	0	0.71	<0.001
FoxP3+Helios-	0.60	0.011	0	0.32	0.018	0.62	0.008	0	0.42	0.005
CD45RA+CD45RO-	0.82	<0.001	0	0.71	<0.001	0.28	0.277	0	0.11	0.199
CD45RA+CD45RO-FoxP3+Helios+	0.31	0.233	0	0.75	<0.001	0.05	0.0861	0	<0.01	0.998
CD45RA-CD45RO+FoxP3+Helios+	0.42	0.094	0	0.26	0.037	0.59	0.134	0	0.61	<0.001
CD8+FoxP3+	0.34	0.3448	1	0.11	0.193	0.69	0.002	0	0.39	0.007
CD8+Helios+	0.23	0.374	0	0.09	0.252	0.14	0.59	0	<0.01	0.915
MDSC, 20 MZ and 13 DZ pairs										
CD14+	0.51	0.021	0	0.33	0.008	−0.23	0.448	0	0.01	0.720
Lin-	0.39	0.092	0	0.63	<0.001	0.09	0.78	0	0.22	0.106
Lin-CD14-	0.53	0.016	0	0.65	<0.001	0.31	0.306	0	0.31	0.047
Lin-CD14-CD15+	0.34	0.1414	0	0.14	0.096	0.82	0.001	0	0.45	0.012
Lin-CD14-CD15+CD11b+	0.14	0.707	0	<0.01	0.876	0.49	0.216	0	0.33	0.140
Lin-CD16-CD14+	0.45	0.047	0	0.38	0.004	0.10	0.734	0	<0.01	0.846
Lin-CD16-CD14+HLA-DR+	0.39	0.088	0	0.25	0.025	−0.07	0.817	0	<0.01	0.961
Lin-CD16+CD14+	0.49	0.029	0	0.14	0.077	0.178	0.583	0	0.01	0.799
Lin-CD16+CD14+HLA-DR+	0.49	0.030	0	0.19	0.056	0.07	0.831	0	<0.01	0.952
Lin-CD16+CD14low	0.79	<0.001	2	0.51	0.001	0.51	0.074	0	0.24	0.088
Lin-CD16+CD14lowHLA-DR+	0.78	<0.001	0	0.65	<0.001	0.l15	0.629	0	0.01	0.751
Lin-CD16-CD14-	0.26	0.260	0	0.09	0.194	0.80	0.001	0	0.50	0.007
Lin-CD14+HLA-DR-	0.82	<0.001	3	0.72	<0.001	0.77	0.002	1	0.58	0.002
Lin-CD14+HLA-DR+	0.43	0.056	0	0.20	0.047	−0.13	0.668	0	0.01	0.762
Lin-CD14-HLA-DR-	0.70	0.001	0	0.45	0.001	0.10	0.748	0	<0.01	0.897
Lin-CD14-HLA-DR+	0.78	<0.001	0	0.71	<0.001	0.25	0.42	0	0.27	0.070

^1^ Nonparametric = nonparametric Spearman correlation. ^2^ Lin. = linear regression. Sr Spearman r; outlier detected by GraphPad Prism.

## Data Availability

The data presented in this study are available on request from the corresponding author.

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
