# Peer review of "Impact of Cytomegalovirus Infection and Genetic Background on the Frequencies of Peripheral Blood Suppressor Cells in Human Twins"

_pathogens, 2021, doi:10.3390/pathogens10080963_

Round 1

Reviewer 1 Report

Goldeck and co-authors present a descriptive analysis of subsets of suppressor cells in circulation (peripheral blood) from CMV seropositive individuals as compared with seronegative controls. The authors specifically address genetic heterogeneity using twin pairs. This is a straightforward pilot study and the authors appropriately describe the limitations of working with small, unique human cohorts. My comments are clarification questions regarding the data as presented here.

Comments:

Line 80: “this cohort”. The cohort of people used in this study is key to the conclusions drawn by the authors. The cohort (its components, background, participant information, numbers/group, etc) should be introduced briefly in the results section (lacking) and in detail in the methods (sufficient as is). The descriptions of the cohort are either scattered, i.e. overall N is on line 205; or vague, i.e. described as “middle-aged”. It would be helpful if the authors were more direct and precise. At least a brief description including #, average age, reference to the Danish Twin Registry and reference to Table S2 is important in the main paper.

Line 106: When were participants tested for antibody response? Was this done in a parallel sample to the lymphocytes used for flow cytometry analysis? Or was this done on previously banked samples. I feel this is important information as it could influence the conclusions.

Line 113: What do the authors mean by “grouped according to the number of different CMV molecules their antibodies bound”. Does this mean grouping individuals by if they are seropositive to 1 (say IE) vs 3 (say IE and p65 and gB)?

The authors should show their gating strategy for identification of cellular subsets in a supplemental figure. (Although this is well described in the methods – thank you).

The data described in S2 and S3 are significant and contribute to their overall conclusions. Is this not worth including in the main manuscript?

Minor comments:

Line 69: MZ and DZ should be defined

Discussion: when referring to the data it might be helpful to also refer back to the appropriate figure #s.

Reviewer 2 Report

The authors present here a pilot study on CMV-induced effects on peripheral blood leukocyte numbers. They utilize a sex-matched twins cohort which elegantly excludes some major confounders and allows a screen for robust phenotypes. The authors state themselves that they present here more or less "negative data" which is an honest statement that should be appreciated for the revision process. The study design and data analysis are of value for readers as flow cytometry approaches for immune phenotyping and screens for phenotype to disease associations are performed in many laboratories. Thus, twin studies are very helpful for interpretation of small cohort studies where the "genetic background" cannot be excluded as a potential confounder.

major point:

Given a high inter- and intra- individual variation of absolute leukoocyte numbers in peripheral blood (day time of blood withdrawal? medication? other infections? other diseases such as autoimmune disorders?...) it is absolutely nescessary to perform such immune phenotyping studies with absolute cell numbers per volume instead of relative distributions. Can you please provide this data?

minor point: I have no doubts that the authors are highly experienced in performing FACS data analysis. However, a representative gating strategy would allow less-experienced readers to follow the technical approaches
